# Possible Use of Linear Echobronchoscope for Diagnosis of Peripheral Pulmonary Nodules

**DOI:** 10.3390/diagnostics13142393

**Published:** 2023-07-17

**Authors:** Lina Zuccatosta, Francesca Gonnelli, Gianmarco Gasparini, Arianna Duro, Francesca Barbisan, Gaia Goteri, Giulia Veronesi, Rocco Trisolini, Stefano Gasparini

**Affiliations:** 1Pulmonary Diseases Unit, Azienda Ospedaliero Universitaria delle Marche, 60126 Ancona, Italy; francesca.gonnelli@ospedaliriuniti.marche.it (F.G.); s.gasparini52@gmail.com (S.G.); 2Department of Biomedical Sciences and Public Health, Polytechnic University of Marche Region, 60126 Ancona, Italy; gianmarco.gasparini@gmail.com (G.G.); ariannaduro@gmail.com (A.D.); 3Pathological Anatomy Institute, Polytechnic University of Marche Region, 60126 Ancona, Italy; francesca.barbisan@ospedaliriuniti.marche.it (F.B.); g.goteri@staff.univpm.it (G.G.); 4School of Medicine and Surgery, Vita-Salute San Raffaele University, 20132 Milan, Italy; veronesi.giulia@hsr.it; 5Department of Thoracic Surgery, IRCCS San Raffaele Scientific Institute, 20132 Milan, Italy; 6Interventional Pulmonology Unit, Policlinico Universitario Agostino Gemelli, 00168 Rome, Italy; rocco.trisolini@policlinicogemelli.it

**Keywords:** peripheral pulmonary nodule, linear echobronchoscope, EBUS-TBNA, transbronchial needle aspiration, transbronchial approach to pulmonary nodules

## Abstract

Echobronchoscope-guided transbronchial needle aspiration (EBUS-TBNA) is mainly used as the transbronchial approach to hilar/mediastinal lymph nodes or lesions, for diagnostic or staging purposes. Moreover, the role of linear EBUS-TBNA as a diagnostic tool for central intrapulmonary lesions adjacent to the trachea or the major bronchi is also well established. However, since the tip of the ultrasound probe at the distal end of the echobronchoscope is very thin, it can be wedged through smaller peripheral bronchi, reaching the distal parenchyma and allowing for peripheral pulmonary lesion sampling. The main aim of this retrospective study was to evaluate the diagnostic yield and the safety of EBUS-TBNA in the diagnosis of pulmonary peripheral nodules. The database of the Interventional Pulmonology Unit of Azienda Ospedaliero-Universitaria delle Marche (Ancona, Italy) was evaluated to identify peripheral pulmonary nodules approached by EBUS-TBNA. Thirty patients with a single peripheral pulmonary nodule located peripherally to the subsegmental bronchi of the lower lobes and adjacent to a small bronchus greater than 3 mm in diameter were included in this study. The nodule was visible using endoscopic ultrasound in 28 patients and the diagnosis was obtained via EBUS-TBNA in 26 cases (12 adenocarcinoma, 5 typical carcinoid tumors, 4 hamartoma and 5 metastatic lesions). The diagnostic yield was 86.6% for all 30 patients and 92.8% if only the 28 patients in which the lesion was visualized via echobronchoscopy were considered. No relevant adverse events were observed. We conclude that EBUS-TBNA may be an effective and safe option to sample pulmonary peripheral nodules in selected patients with lower lobe peripheral pulmonary lesions adjacent to small bronchi greater than 3 mm in diameter and reachable with the EBUS-TBNA probe.

## 1. Introduction

Peripheral pulmonary nodules (PPNs) represent a frequent incidental finding on a chest computed tomography scan (CT). With the increase in lung cancer screening trials using low dose CT, the incidence of PPNs has also risen, reaching up to 33% in the high-risk smoker population [1,2]. Although most incidentally identified PPNs are benign, for nodules larger than 1 cm, the risk of its malignant nature increases significantly, and surgical removal without prior histological diagnosis is carried out less frequently [3].

A PPN can be approached for diagnostic purposes using CT-guided transthoracic needle biopsy or with guided bronchoscopy sampling. While transthoracic needle biopsy provides a better sensitivity, it is associated with a higher incidence of complications, particularly pneumothorax [4]. In recent years, in addition to the traditional fluoroscopy, new guidance systems have been developed and used along with bronchoscopy, to improve the diagnostic yield of the transbronchial approach to PPNs. Virtual bronchoscopy, electromagnetic navigation systems, radial ultrasound miniprobes, cone beam CT and more recently, robotic bronchoscopy have been employed, alone or in combination, and have been able to increase the sensitivity of guided bronchoscopy for the diagnosis of PPNs to values as high as 80–85% [5].

The convex-probe echobronchoscope is usually not mentioned among the instruments used for sampling PPNs, as it is mainly used for the transtracheal or transbronchial approach to hilar/mediastinal lymph nodes or masses, for both diagnostic and staging purposes [6,7]. However, endobronchial ultrasound-guided transbronchial needle aspiration (EBUS-TBNA) has also been safely and successfully employed for the diagnosis of bronchoscopically invisible intrapulmonary lesions located adjacent to the central airways [8]. Moreover, in everyday clinical practice, even pulmonary lesions located beyond the subsegmental bronchi are sometimes identified with EBUS-TBNA. This is possible because the ultrasound transducer (6.9 mm in its greater diameter) tapers towards its tip, where the scope ends in a pedicel that is 3 mm. By wedging this pedicel into the bronchus, it is possible to gently enlarge the airway lumen allowing for the penetration of the thinner part of the transducer. Since the ultrasound field of vision has a conical shape, the above maneuver may be sufficient to visualize the nodule and sample it under real-time visualization. However, this technique is only feasible for nodules located in the lower lobes, as the progression of the echoendoscope is not possible or very difficult in the segmental bronchi of the upper lobes and of the middle lobe.

To our knowledge, the possible role of EBUS-TBNA for diagnosing PPNs was not previously assessed.

The aim of the present study was to evaluate the feasibility and the diagnostic yield of EBUS-TBNA for the diagnosis of bronchoscopically invisible peripheral pulmonary nodules located in the lower lobes.

Secondary endpoints included evaluating the rate of complications and the adequacy of the samples for lung cancer molecular profiling.

## 2. Materials and Methods

### 2.1. Study Design and Patient Selection

This was a retrospective study undertaken at the Interventional Pulmonology Service of the Azienda Ospedaliero-Universitaria delle Marche (Ancona, Italy) between January 2018 and February 2023. The EBUS-TBNA database was reviewed, and the EBUS-TBNA procedures performed for diagnosis of PPNs ≤ 3 cm in diameter were identified. Typically, patients with a PPN were submitted to a convex-probe EBUS evaluation in the presence of the following criteria:(a)pulmonary nodule located in the lower lobes, in an area adjacent to the subsegmental or a more distal bronchus;(b)evidence at CT scan of an airway ≥ 3 mm in diameter adjacent or close (≤1 cm) to the nodule;(c)absence of endobronchial abnormalities visible based on conventional flexible bronchoscopy.

### 2.2. Procedure

In all patients, EBUS-TBNA was performed after the nebulization of topical anesthesia (lidocaine 2%) by experienced bronchoscopists (LZ, SG) under moderate sedation (midazolam 0.035 mg/kg + fentanyl 0.00035 mg/kg). During the procedure, arterial oxygen saturation and electrocardiographic tracing were systematically monitored.

Following a conventional bronchoscopy aimed at ruling out the presence of visible endobronchial lesions, the echobronchoscope (Olympus BF-UC-180F, Olympus Corporation, Tokyo, Japan) was introduced into the airways and brought into the subsegmental bronchus thought to lead to the lesion based on the evaluation of the CT axial and multiplanar reconstructions. The tip of the scope was then wedged into the airway, and the instrument was rotated until the nodule was ultrasonographically visible. If the lesion was not immediately identified, the position of the echobronchoscope with regard to the nodule location was assessed using fluoroscopy, and an attempt at again directing the scope towards the nodule was made.

If the nodule was identified, transbronchial needle aspiration was performed under real-time ultrasound guidance using a 22-gauge needle (Olympus NA-201SX-4022).

After the procedure, the patients were monitored in a recovery room for 2 h to capture any adverse effect before being discharged from the Interventional Pulmonology Service.

### 2.3. Sample Management

Four needle passes were carried out for each patient. The sample retrieved with the first needle pass was smeared on clean glass slides. One slide was immediately stained using a rapid method (Hemacolor Merck; Darmastadt, Germany) for rapid-on-site evaluation (ROSE), and the other slides were fixed in 95% ethanol and later stained in the Pathology Lab with the Papanicolau method for definitive cytological evaluation. Then, the other 3 needle passes were carried out, and the material was flushed in 10% neutral-buffered formalin for cell-block.

In the case of a final diagnosis of pulmonary adenocarcinoma, cytological smears were used for DNA extraction (QIA amp DNA mini kit). Smears were considered adequate for tumor genotyping if at least 500 neoplastic cells were present with a tumor vital cellularity equal or above 50%. Mutational analysis of the 10 genes commonly tested in the setting of non-small cell lung cancer (EGFR, KRAS, BRAF, PIK3CA, NRAS, ALK, ERBB2, DDR2, MAP2K1 and RET) was performed via MALDI-TOF Mass Spectrometry (MassARRAY, Agena Bioscience, San Diego, CA, USA) using the Myriapod Lung Status Kit (Diatech Pharmacogenetics). Cell blocks were utilized for the immunohistochemical predictive markers ALK (D5F3 CDx assay on platform BenchMark ULTRA Ventana Medical Systems Inc., Basel, Switzerland), ROS1 (Ventana SP384 Rabbit Monoclonal Primary Antibody on platform BenchMark ULTRA Ventana Medical Systems Inc., Basel, Switzerland) and PD-L1 (PD-L1 IHC 22C3 pharmDx for Autostainer Link 48-Agilent) [9].

### 2.4. Endopoints and Statistical Analysis

The primary endpoint for this study was to assess the feasibility and the diagnostic yield of EBUS-TBNA for the diagnosis of pulmonary nodules located in the periphery of lower lobes. Secondary endpoints included the following: (i) the evaluation of the complication rate, and (ii) the adequacy of the samples for the molecular profiling of lung cancer.

The diagnostic yield was calculated as the number of cases in which EBUS-TBNA provided a diagnosis relative to the total number of cases with a PPN for whom EBUS-TBNA was performed for diagnostic purposes.

To evaluate if the size of the lesion may influence the diagnostic yield, a subgroup analysis according to the diameter of the nodule (≤15 mm; >15 mm) was carried out using Fisher’s Exact Test.

### 2.5. Ethical Aspects

This retrospective study was conducted in accordance with the Declaration of Helsinki and approved by the Ethics Committee of Marche Region (CERM:172/2023). Due to the retrospective nature of the study and since data were de-identified, the need for patient informed consent was waived.

## 3. Results

From January 2018 to February 2023, 844 EBUS-TBNAs were performed at the Interventional Pulmonology Service of the “Azienda Ospedaliero-Universitaria delle Marche”, Ancona, Italy.

Here, 739 procedures were performed to sample hilar/mediastinal lymph nodes or masses for diagnostic and/or staging procedures; 75 EBUS-TBNAs were performed for diagnosing intrapulmonary central lesions adjacent to the trachea or main bronchi (*n* = 57) or peripheral masses greater than 3 cm (*n* = 18). In the remaining 30 patients, who represent the cohort of the present study, a bronchoscopically invisible peripheral nodule ≤3 cm, close to a subsegmental or more distal bronchus, was sampled.

The baseline characteristics of patients and lesions are reported in Table 1.

There were 16 males and 14 females (mean age 66 yrs; range: 18–84). The mean diameter of the lesions was 20.3 mm (min 8; max 30). Twelve patients had a PPN ≤ 15 mm. Seventeen nodules were located in the right and 13 in the left lung.

In 19 cases (63.3%%), the lesion was directly identified ultrasonographically without the need for any other imaging guidance, whereas in 9 cases (30%) the localization of the nodule was achieved with the help of fluoroscopy, which allowed for the identification of the correct direction in which the bronchoscope had to be repositioned and wedged. In the remaining two cases, the lesion could not be visualized by the echobroncoscope despite the use of fluoroscopy; of these, one was in the medial segment of the right lower lobe and the other was in the lateral segment of the left lower lobe, and both lesions had a diameter of 20 mm.

We did not notice any difference in the relationship between the lesion and the airway (close or adjacent) in cases in which fluoroscopy was used. The reason why the nodule was not immediately visualized ultrasonographically was the placement of the instrument in a subsegmental bronchus, not close or adjacent to the lesion.

Of the 28 nodules identified by the echobronchoscope, a needle aspiration was possible in all cases (93.3% feasibility) and the definitive diagnosis was obtained in 26 (Table 2). The diagnostic yield for a tissue diagnosis was 26/30 (86.6%) if we considered the whole patient population (worst case scenario) or 26/28 (92.8%) if we included in the analysis only patients in whom the lesion was ultrasonographically visualized (best case scenario).

In 20 cases, the nodule was closely adjacent to the bronchus, and in 10 cases, it was near the bronchus with a maximum distance of 10 mm.

In the two cases in which the lesion could not be visualized via EBUS, the diagnosis was obtained in one case with bronchoscopy guided by radial EBUS and fluoroscopy (pulmonary adenocarcinoma) and in the other case, in the surgical resection specimen (carcinoid tumor).

In the remaining two patients, the lesion was correctly visualized and sampled, but the EBUS-TBNA failed to provide a definitive diagnosis, showing non-specific inflammatory material. These two patients are still under clinical and radiological follow-up.

The diagnostic yield of EBUS-TBNA was not significantly different for a nodules ≤ 15 mm or >15 mm (11/12, 91.6% versus 15/18, 83.3%; *p* = 0.63).

In all patients, tissue cores for cell block evaluation were obtained. In the 12 patients finally diagnosed with pulmonary adenocarcinoma, the EBUS-TBNA specimens allowed for a complete molecular profiling in all cases.

In two cases, CT scan showed lymph node enlargement, but not an increase in PET uptake, in station 11 (short axis diameter = 1.7 and 1.9 cm respectively). In these two cases, systematic endosonographic staging was performed, but no evidence of lymph node metastases was found during the histological examination.

No serious adverse events were reported. In two patients, mild bleeding occurred, which was easily controlled using bronchoscopic suction without the need of supplementary therapeutical measures and without clinical consequences.

Figure 1 and Figure 2 show the CT findings and the paired ultrasonographic appearance of the pulmonary nodule in some of the patients included in the present series.

## 4. Discussion 

The transbronchial diagnostic approach to peripheral pulmonary nodules currently remains a challenge for interventional pulmonologists. Despite the availability of new techniques and advanced guidance systems (e.g., ultrathin bronchoscopes, virtual bronchoscopy navigation, electromagnetic navigation, cone beam CT, robotic bronchoscopy) that allow for the visualization of even small nodules in most cases, the diagnostic sensitivity of guided bronchoscopy does not exceed 80% in most series, as very recently confirmed by a comprehensive systematic review with a meta-analysis [5,10].

Of the above guidance systems, the radial ultrasound miniprobes (rEBUS), is one of the most used, alone or in association with other technologies [11,12]. rEBUS employs a rotating ultrasound transducer located at the end of the mini probe, able to provide a 360° view of the peribronchial structures. The miniprobe is introduced through the working channel of the bronchoscope and thrust into the peripheral airways until a characteristic ultrasound signal produced by a solid or subsolid lesion, markedly different from the “snowstorm” appearance produced by the normal parenchyma, is visualized. According to the systematic reviews with meta-analysis, the sensitivity of bronchoscopy guided by rEBUS for the diagnosis of PPNs ranges from 69 to 73% [13,14,15,16]. In a study on 200 patients with peripheral lung lesions approached for biopsy purposes with rEBUS, Eom JS et al. [17] obtained an overall diagnostic yield of 73%, but the sensitivity was significantly different among patients whose lesions had a mean diameter < 20 mm (46.8%) versus lesions greater than 20 mm (80.8%). Furthermore, the diagnostic yield of cases in which the ultrasound probe was located within the lesion was significantly higher than that in cases in which the probe was adjacent to the lesion (80% versus 58%). The failures associated with this procedure are related to one or more of the following limitations: (a) bronchoscopy guided by rEBUS does not allow for real-time sampling; (b) the diagnostic yield is significantly lower when the lesion grows outside the airway (eccentric location with respect to the radial probe) than when it grows at least in part in the airway (concentric location with respect to the radial probe) (5). The convex-probe EBUS helps to overcome both the above limitations, as it allows for the real-time visualization of the needle within the lesion during sampling, regardless of whether the nodule has a peri-bronchial or endobronchial growth pattern.

The use of EBUS-TBNA for sampling central parenchymal lesions that do not compress or infiltrate the tracheobronchial mucosa, but are adjacent to the airways, has been already described in several papers. Tournoy KG et al. [18] retrospectively evaluated, with EBUS-TBNA, 60 patients with an intrapulmonary mass located in the inner third of the hemithorax on chest CT-imaging. The authors were able to identify the lesion with ultrasound in all cases, but the sampling was performed in 58. A definitive diagnosis was obtained using the EBUS-TBNA specimens in 46 patients, with a sensitivity for lung cancer of 82%. In a retrospective multicenter study [8], Kuijvenhoven et al. evaluated 163 bronchoscopically invisible centrally located intrapulmonary tumors near or adjacent to the airways up to the segmental bronchi. Lung tumor sampling via EBUS was feasible in 145 patients (89%). In four cases, the lung tumor was not found with EBUS, and in 14 cases, sampling was not performed for different reasons, mainly for the interposition of large vessels between the airways and the lesion. A definitive diagnosis was achieved using EBUS-TBNA in 136 patients with a diagnostic yield of 94%. Chaiyakul S. investigated the diagnostic performance of convex probe endobronchial ultrasound-guided transbronchial needle aspiration in 175 patients with central intrapulmonary lesions with a diagnostic yield of 90.3% [19]. Nakajima T et al. performed EBUS-TBNA on 35 patients with pulmonary masses located close to the central airways (19 peritracheal and 16 peribronchial lesions) with a sensitivity of 94.1% [20]. Yang H et al. performed EBUS-TBNA on 68 tracheobronchial wall-adjacent intrapulmonary lesions with a sensitivity of 93.4% [21]. However, in all the above mentioned studies, the size of the lesion was not an exclusion criteria and also masses (>3 cm in long axis diameter) were included.

Besides peri-bronchial central lung lesions, peri-esophageal nodules and masses can be sampled with an echobronchoscope by inserting it into the esophagus (EUS-B-FNA). In a multicenter study, Mondoni et al. described the results of EUS-B-FNA with 107 patients with a pulmonary paraesophageal lesion [22]. The diagnostic accuracy was 95.3% in the overall cohort and 95.2% in the 99 patients with a final diagnosis of malignancy. In this study, the mean lesion size was 42.2 mm (range 32–59 mm) and not a single nodule ≤ 3 cm was included in the study cohort.

To the best of our knowledge, the present study is the first one that assesses the possible role of convex-probe EBUS for the diagnosis of peripheral pulmonary lesions < 3 cm in long-axis size. Remarkably, the 92.8% diagnostic yield that we obtained in patients in whom the lesion could be visualized ultrasonographically is even higher than that reported in the literature with the use of the more advanced technologies, such as electromagnetic navigation bronchoscopy (EMNB), cone beam CT or robotic bronchoscopy. The diagnostic yield of EMNB, evaluated in systematic reviews with metanalyses, ranges from 64.9 to 82% [23,24,25,26]. Higher diagnostic yields have been reported with the use of cone beam CT (70–90%), although in most such studies, cone beam CT was used together with other guidance systems, such as EMNB and/or rEBUS, or with the use of ultrathin bronchoscopes [27,28,29]. More recently, the introduction of robotic bronchoscopy has further improved the diagnostic success of the transbronchial approach to the diagnosis of peripheral pulmonary nodules, with yields ranging from 74 to 92% [30,31,32], but with increased procedure complexity and costs. Echobronchoscopes are currently available in most interventional pulmonology units, many practitioners have quite extensive experience with EBUS-TBNA of lymph nodes and centrally located pulmonary lesions and the procedure, not requiring disposable materials, is much less costly than a guided bronchoscopy, especially if advanced guidance systems are used. This altogether makes EBUS-TBNA an appealing procedure for diagnosing PPNs with the characteristics of those included in our study.

In our series, only nodules ≤ 3 cm were included, but EBUS-TBNA can obviously be used for larger lesions located in the lung periphery. Interestingly, the lesion size did not influence the likelihood of obtaining the final diagnosis in our series, a result suggesting that once the lesion is visualized ultrasonographically, the chances of retrieving diagnostic tissue is independent of its size. Indeed, we could visualize and sample even 8 or 9 mm nodules (cases B and C of Figure 2), which were finally diagnosed first as a pulmonary metastasis of urothelial cell cancer and second as a carcinoid tumor. We hypothesize that the reason for the failure to visualize the nodule in the two cases in whom this happened was related to a tributary bronchus too angled to allow for the insertion and progression of the tip of the scope.

The major limitation of EBUS-TBNA in sampling PPNs is the possibility of using this technique almost exclusively for lesions localized in the lower lobes. Due to the outer diameter of the instrument and to the anatomy of the bronchial tree, in fact, it is not possible to thrust the echobronchoscope in the subsegmental bronchi of the upper lobes.

We tried to insert the echobronchoscope in the subsegmental bronchi of the middle lobe, but we succeeded in making it progress, at most, at the level of the segmental bronchi. The same difficulty of making the instrument progress into the subsegmental bronchi of the middle lobe could be predicated for the superior bronchus n.6 of the lower lobe. In our series, no one case was located in the superior segment of lower lobes.

Recently, thinner echobronchoscopes have been introduced into the market [33], with an outer diameter of 6.6 mm, a shorter and thinner ultrasound transducer of 25 mm and a 160° angulation capability. The possibility of accessing subsegmental bronchi of the upper lobes with this slim EBUS scope has not yet been tested. We hypothesize that with the slim echobronchoscope, the access to more peripheral nodules and, at least, a deeper progression in the subsegmental bronchi of the middle lobe and of the superior segment of the lower lobes could be possible.

Another limitation to the successful use of EBUS-TBNA in diagnosing PPNs is the need for a bronchus ≥ 3 mm adjacent to the lesion, although this airway does not need to be in close contact with the nodule. Indeed, in our experience (as reported in case B of Figure 2) a distance of up to 1 cm can be markedly shortened, if not completely eliminated, by bending the tip of the scope towards the lesion, a maneuver that usually brings the ultrasound probe in contact or very close to the nodule.

As already reported in our previous experiences [9,34], the material obtained via needle aspiration can be adequate for the complete genotyping of the peripheral lung tumors, if correctly managed. By using both smeared material and cell blocks, complete molecular profiling, including Programmed Death Ligand 1 (PDL1), was obtained in all cases of lung adenocarcinoma.

Only two minor bleeding cases, easily managed using a bronchoscope and without clinical consequences, were noticed in our study. This confirms the safety of EBUS-TBNA both overall and more specifically for diagnosing PPNs.

The results of this study should be interpreted in the context of some limitations, in particular the retrospective design and the small sample size. For instance, important procedural details, such as the duration of the exam, were not systematically recorded. In addition, we set the inclusion criteria based on our personal clinical experience. A prospective evaluation of the role of the “thin” convex probe EBUS for the diagnosis of any PPNs located in the lower and middle lobes, ideally not setting “strict” inclusion criteria regarding the size of the bronchus adjacent to the lesion, could help to more reliably define feasibility, the diagnostic yield and predictors of a successful procedure. Finally, the study was carried out at a high-volume center by experienced operators; as a consequence, the reproducibility of the results in settings with less experience is unclear.

## 5. Conclusions

In conclusion, this preliminary experience demonstrates that in patients with peripheral pulmonary nodules located in the lung lower lobes and adjacent or close to a bronchus ≥ 3 mm in diameter, EBUS-TBNA can be a safe and effective means of sampling, regardless of the lesion size.

Knowledge of this possibility would offer interventional pulmonologists a safe, cheap and successful alternative to more advanced guided bronchoscopy systems for diagnosing PPNs with the above-described characteristics.

## Figures and Tables

**Figure 1 diagnostics-13-02393-f001:**
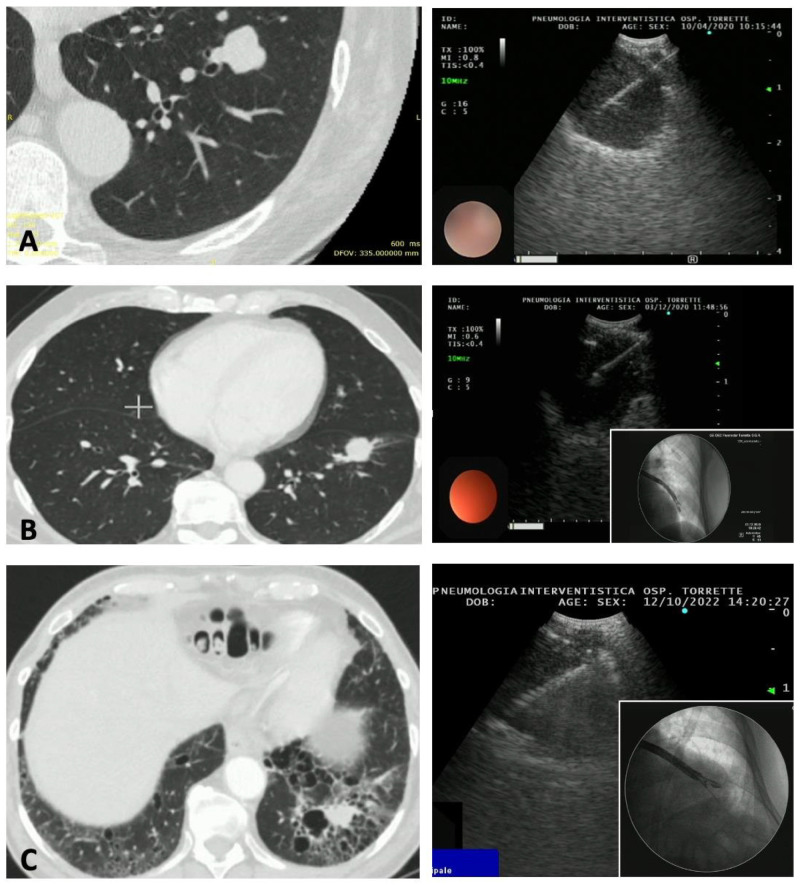
Three cases of peripheral pulmonary nodules of the left lung approached with EBUS-TBNA for diagnostic purposes. (**A**) Carcinoid tumor of the lateral segment of the left lower lobe. (**B**) Adenocarcinoma of the lateral segment of the left lower lobe; in the inset, a fluoroscopic image was obtained during the procedure, showing the echobronchoscope close to the nodule. (**C**) Adenocarcinoma of the posterior segment of the left lower lobe. Emphysematous bullae are present around the nodule, making a percutaneous approach at high risk for pneumothorax.

**Figure 2 diagnostics-13-02393-f002:**
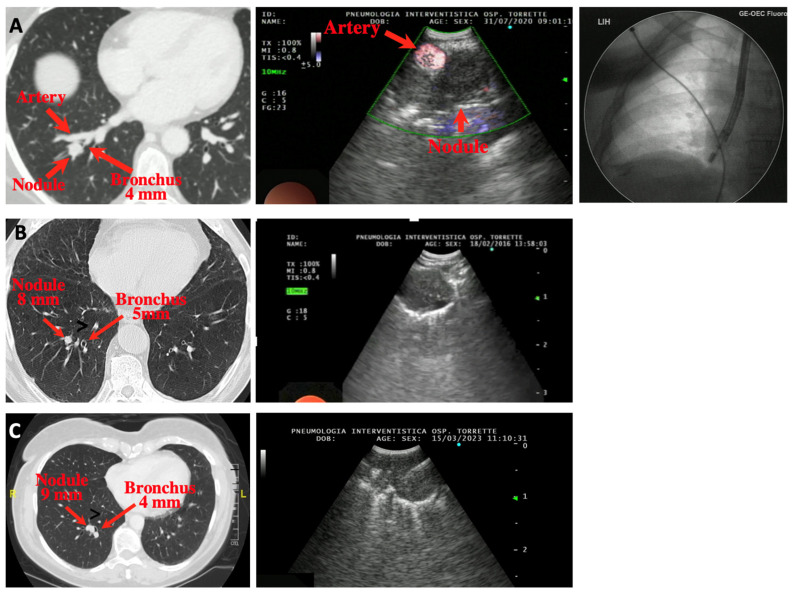
Three cases of peripheral pulmonary nodules of the right lung approached using EBUS-TBNA. (**A**) Paired CT, ultrasound and fluoroscopic images of a case of adenocarcinoma of the posterior segment of the left right lower lobe. In the ultrasound image, a large branch of the pulmonary artery lying very close to the nodule is shown using the Power Doppler function. (**B**) A small nodule (8 mm) of the posterior segment of the right lower lobe (metastasis of urothelial cancer). The nodule was successfully sampled, despite a 1 cm distance from the closest bronchus, by bending and thrusting the tip of the scope towards the nodule. (**C**) A small nodule (9 mm) of the posterior segment of the right lower lobe (carcinoid tumor). In the ultrasound image, the needle inside the lesion is visible.

**Table 1 diagnostics-13-02393-t001:** Baseline characteristics of patients with a bronchoscopically invisible peripheral pulmonary nodule, who underwent EBUS-TBNA for diagnostic purposes.

**Gender**	
Male *n* (%)	16 (53.3)
Female *n* (%)	14 (46.7)
**Age**, year. Mean (range)	66 (18–84)
**Size of the lesion**, mm. Mean (range)	20.3 (8–30)
**Location of the PPNs (*n*)**	
**Right lower lobe**	
Medial segment	3
Anterior segment	3
Lateral segment	5
Posterior segment	6
**Left lower lobe**	
Anteromedial segment	6
Lateral segment	3
Posterior segment	4

**Table 2 diagnostics-13-02393-t002:** Diagnostic yield of EBUS-TBNA in 30 patients with a bronchoscopically invisible peripheral pulmonary nodule.

Lung Adenocarcinoma	12
Typical carcinoid	5
Hamartoma	4
Metastasis	5
-renal cell carcinoma	2
-breast cancer	1
-melanoma	1
-urothelial cell carcinoma	1
Non diagnostic	2
Not visualized via EBUS	2

## Data Availability

The data that support the findings of this study are available from the corresponding author upon reasonable request.

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
