# Peer review of "Possible Use of Linear Echobronchoscope for Diagnosis of Peripheral Pulmonary Nodules"

_diagnostics, 2023, doi:10.3390/diagnostics13142393_

Round 1
Reviewer 1 Report
Review comments
Possible Use of Linear Echobronchoscope for Diagnosis of Peripheral Pulmonary Nodules
EBUS-TBNA is a powerful diagnostic tool for various types of lung tumors. This study provides a new diagnostic technique using a convex probe EBUS-bronchoscope. It must be exciting for many bronchoscopists or pulmonology physicians.
My only minor comments are as follows;
1. In the introduction, the authors described “the tip of the ultrasound transducer at the distal end of the echobronchoscope is 3mm”. Is this correct? The edge of the transducer of UC180F (the authors used in this study) would be more than 5mm. Even if the authors used UC190F (not UC 180F), the tip of a transducer might be more than 3mm. Please confirm.
2. There were two cases where EBUS could not visualize the lesion. The authors should present the reason for failure (the responsible bronchus was too small? or rigid?).
Author Response
Reviewer #1
EBUS-TBNA is a powerful diagnostic tool for various types of lung tumors. This study provides a new diagnostic technique using a convex probe EBUS-bronchoscope. It must be exciting for many bronchoscopists or pulmonology physicians.
The Authors thank the reviewer for the favorable and helpful comments aimed at improving the manuscript.
My only minor comments are as follows;
- In the introduction, the authors described “the tip of the ultrasound transducer at the distal end of the echobronchoscope is 3mm”. Is this correct? The edge of the transducer of UC180F (the authors used in this study) would be more than 5mm. Even if the authors used UC190F (not UC 180F), the tip of a transducer might be more than 3mm. Please confirm.
Thank you for this comment that allow us to specify better the technique.
The reviewer is correct as the size of the UC180F echobronchoscope transducer is 6.9 mm in its greater diameter. However, the convex shape of the transducer tapers towards its tip, where the scope ends in a pedicel that is 3 mm. By wedging this pedicel in the bronchus it is possible to gently enlarge the airway lumen allowing the penetration of the thinner part of the transducer. Since the ultrasound field of vision has a conical shape, the above maneuver may be sufficient to visualize the nodule.
The following sentence has been added in the introduction:
“the ultrasound transducer (6.9 mm in its greater diameter) tapers towards its tip, where the scope ends in a pedicel that is 3 mm. By wedging this pedicel in the bronchus, it is possible to gently enlarge the airway lumen allowing the penetration of the thinner part of the transducer. Since the ultrasound field of vision has a conical shape, the above maneuver may be sufficient to visualize the nodule and sample it under real-time visualization”.
- There were two cases where EBUS could not visualize the lesion. The authors should present the reason for failure (the responsible bronchus was too small? or rigid?).
Thank you for this comment.
We hypothesize that the reason for failure was a tributary bronchus too angled to allow for the echobronchoscope progression.
The following sentence has been added in the discussion:
“We hypothesize that the reason for failure to visualize the nodule in the two cases in whom this happened was related to a tributary bronchus too angled to allow the insertion and progression of the tip of the scope”.
Reviewer 2 Report
This is a retrospective study evaluating the diagnostic yield and the safety of EBUS-TBNA in the diagnosis of pulmonary peripheral nodules. The topic of this article is of interest. The manuscript is generally well-written, and can be improved by addressing some points in the revision.
1. As mentioned by the authors, this is a retrospective study with small sample size. It could be hard to trace back all the the procedure details, for instance, the number of punctures, were tissue cores acquired in all cases (eespecially the non-diagnostic ones). What were the pathology findings for the non-diagnostic ones. Clarification is needed for non-diagnostic.
2. What is the average procedure duration and average time for lesion localization? As mentioned in this study, 9 (30%) PPLs was localized with the guidance of fluoroscopy. Were the lesions reached under fluoroscopy tend to be close to bronchus, rather than adjacent to bronchus? What is the additional time for fluoroscopy?
3. In the results part, the author mentioned “From January 2018 to February 2023, 844 EBUS-TBNA were performed, 739 procedures were performed to sample hilar/mediastinal lymph nodes or masses for diagnostic and/or staging procedures. 75 EBUS-TBNA were done for diagnosing of intrapulmonary central lesions adjacent to the trachea or main bronchi. In the remaining 30 patients, who represent the cohort of the present study, a bronchoscopically-invisible peripheral nodule ≤3 cm, close to a subsegmental or more distal bronchus, was sampled.” All the remaining 30 patients happens to have lesions ≤3 cm? No PPLs larger than 3cm were diagnosed by EBUS-TBNA? Was the EBUS-TBNA procedure for PPLs restricted to for lesions<3cm in your center or just written mistake?
4. Did any patients have enlarged hilar or mediastinal lymph nodes? Was concomitant lymph node sampling performed in any patients?
Author Response
Reviewer #2
This is a retrospective study evaluating the diagnostic yield and the safety of EBUS-TBNA in the diagnosis of pulmonary peripheral nodules. The topic of this article is of interest. The manuscript is generally well-written, and can be improved by addressing some points in the revision.
The Authors thank the reviewer for the favorable and helpful comments aimed at improving the manuscript.
- As mentioned by the authors, this is a retrospective study with small sample size. It could be hard to trace back all the procedure details, for instance, the number of punctures, were tissue cores acquired in all cases (especially the non-diagnostic ones). What were the pathology findings for the non-diagnostic ones. Clarification is needed for non-diagnostic.
Thank you for this comment that allow us to better specify the procedure details. For each case, we perform a first needle pass and the sample is smeared on a glass for cytological evaluation. Following the first needle pass, three further punctures are performed and the sample is flushed in formalin for cell-block preparation. This standard procedure was performed in all cases and the tissue core for cell block was obtained in all patients.
We specified this point in the results section, adding the sentence “In all patients tissue core for cell block evaluation was obtained”
The pathological findings of non-diagnostic EBUS-TBNA was in both cases represented by nonspecific inflammatory material. These two cases are still under follow-up and we currently don’t know whether the sample was a false negative for malignancy. We specified this point adding this information in the Result section.
- What is the average procedure duration and average time for lesion localization? As mentioned in this study, 9 (30%) PPLs was localized with the guidance of fluoroscopy. Were the lesions reached under fluoroscopy tend to be close to bronchus, rather than adjacent to bronchus? What is the additional time for fluoroscopy?
Thank you for this comment that allows us to specify that the procedure is very fast if the echobronchoscope is inserted in the correct subsegmental bronchus and the lesion is immediately visualized. As correctly underlined by the reviewer, this was a retrospective study for which not all the procedural details could be retrieved. In particular, the procedural time was not recorded. This has been added to the limitation of the study. This said, the use of fluoroscopy certainly lengthened the procedural time to some extent.
We did not notice any difference in the relationship of the lesion to the bronchi (close or adjacent) in cases where fluoroscopy was used. The reason why the nodule was not immediately visualized was that the bronchoscope was inserted in a wrong subsegmental bronchus, and fluoroscopy helped us to identify the correct direction in which the bronchoscope had to be repositioned and wedged. We added the following sentence in the discussion: “We hypothesize that the reason for failure to visualize the nodule in the two cases in whom this happened was related to a tributary bronchus too angled to allow the insertion and progression of the tip of the scope”.
- In the results part, the author mentioned “From January 2018 to February 2023, 844 EBUS-TBNA were performed, 739 procedures were performed to sample hilar/mediastinal lymph nodes or masses for diagnostic and/or staging procedures. 75 EBUS-TBNA were done for diagnosing of intrapulmonary central lesions adjacent to the trachea or main bronchi. In the remaining 30 patients, who represent the cohort of the present study, a bronchoscopically-invisible peripheral nodule ≤3 cm, close to a subsegmental or more distal bronchus, was sampled.” All the remaining 30 patients happens to have lesions ≤3 cm? No PPLs larger than 3cm were diagnosed by EBUS-TBNA? Was the EBUS-TBNA procedure for PPLs restricted to for lesions<3cm in your center or just written mistake?
Thank you for this comment that underlines a written mistake. In fact, among the 75 cases of pulmonary lesions sampled by EBUS-TBNA, there were also 18 cases of peripheral lesions greater than 3 cm, that we didn’t include in the present study which was focused on nodules (< 3cm).
We have changed accordingly the sentence in : “75 EBUS-TBNA were done for diagnosing of intrapulmonary central lesions adjacent to the trachea or main bronchi (n=57) or peripheral lesions greater than 30 mm close to more peripheral airways (n=18)”
- Did any patients have enlarged hilar or mediastinal lymph nodes? Was concomitant lymph node sampling performed in any patients?
Thank you for this comment that arise a very important point. According to guidelines, systematic endoscopic staging is not indicated for peripheral lesions < 3 cm, if CT scan or PET CT are negative for lymph node involvement. In our series, in only 2 patients CT scan showed a lymph node enlargement at station 11, that was PET negative. EBUS-TBNA was performed in the lymph node in both cases, showing normal lymphoid tissue.
We have added the following sentence in the result section: “In two cases CT scan showed a lymph node enlargement (minor axis = 1.7 and 1.9 cm) in station 11. Even if PET/CT was negative, we performed EBUS-TBNA on these lymph nodes that resulted negative for atypical cells (fragment of normal lymphatic tissue).